# Hydrogen-like Impurity States in β-Ga$_2$O$_3$/(Al$_x$Ga$_{1-x}$)$_2$O$_3$ Core/Shell Nanostructures: Comparison between Nanorods and Nanotubes

**Sihua Ha [1] and Jun Zhu [2],***

[1] College of Sciences, Inner Mongolia University of Technology, Hohhot 010051, China; hasihua_121@163.com
[2] School of Physical Science and Technology, Inner Mongolia University, Hohhot 010021, China
* Correspondence: zhujun@imu.edu.cn

**Abstract:** The binding energy of an off-center hydrogen-like impurity in an ultra-wide band gap β-Ga$_2$O$_3$/(Al$_x$Ga$_{1-x}$)$_2$O$_3$ core/shell nanostructure is studied using a variational method combined with a finite-difference algorithm. Four impurity states with the radial and axial quantum numbers being 0 or 1 in two kinds of core/shell nanostructures, including nanorods and double-walled nanotubes, are taken into account in the numerical calculations. The variation trends in binding energy corresponding to the four impurity states as functions of structural dimension and Al composition differ in nanorods and nanotubes when the impurity moves toward the interface between the Ga$_2$O$_3$ and (Al$_x$Ga$_{1-x}$)$_2$O$_3$ layers. The quantum confinement due to the structural geometry has a considerable influence on the probability density of the impurity states as well as the impurity binding energy. The numerical results will pave the way toward theoretical simulation of the electron states in rapidly developing β-Ga$_2$O$_3$ low-dimensional material systems for optoelectronic device applications.

**Keywords:** binding energy; nanorod; nanotube; quantum confinement





## 1. Introduction

Core–shell nanostructured semiconductors, such as nanorods, nanotubes, and nanodots, are regarded as an intriguing class of materials in electronics and optoelectronics due to their quantum size effect and novel physicochemical features compared with their bulk materials and other low-dimensional structured counterparts [1]. It is known that impurities play a crucial role in semiconductors by adjusting their electric conductivity and optical spectral characteristics [2–4]. To determine the impurity-bound states in core–shell nanostructures, the variational approach [5–13] with different forms of trial wavefunctions containing one [5,6,11], two [6,7,9,12,13], or three [8] variational parameters has been generally adopted for a coupled electron-impurity system with confinement potential. In addition to the variational approach, the finite-difference [14,15] and finite-element [16] numerical methods have also been used to directly solve the Schrödinger equation after the Coulombic potential was decoupled effectively as one. These works show that the binding energy [6–13,15–18], oscillator strength [8], and impurity-induced nonlinear optical properties like the optical absorption coefficient [8,14], refractive index change [8,14], photoionization cross section [9,13,16], and magnetic susceptibility [5–7,10] have great dependence on impurity position, structure dimension, composition, temperature, pressure, as well as external field (electric field, magnetic field, terahertz field, etc.). However, most authors have focused their attention on the impurity states in core–shell nanostructures composed of III-V compound semiconductors, and there still lacks adequate information related to the rapidly developing fourth-generation semiconductor material systems such as Ga$_2$O$_3$ and its sesquioxides.

$Ga_2O_3$ is a promising ultrawide band gap semiconductor with $E_g$ ~4.9 eV for high frequency, high temperature, and high voltage applications in device domains such as power electronics [19,20], solar-blind deep-ultraviolet photodetectors [21,22], gas sensors [23,24], and so on. Among all of the polymorphs, β-$Ga_2O_3$ is the most stable modification. The band alignment between $Ga_2O_3$ and its ternary mixed crystal, $(Al_xGa_{1-x})_2O_3$ or $(In_xGa_{1-x})_2O_3$, is II type with nearly no valence band discontinuity and the conduction band offset can be tuned up to 1.7 eV through alloy composition. The Al composition of the as-grown pure phase $(Al_xGa_{1-x})_2O_3$ can reach $x < 0.71$ for a monoclinic structure and $x > 0.71$ for a corundum structure, in spite of the solid phase miscibility gap between β-$Ga_2O_3$ and α-$Al_2O_3$ [25]. To date, some $Ga_2O_3/(Al_xGa_{1-x})_2O_3$ low-dimensional structures like heterojunction [26,27], quantum well [28,29], and core–shell nanowire [30] have been fabricated using various vacuum techniques. Lyman and Krishnamoorthy [31] recently performed a theoretical investigation of optical intersubband transitions of electrons in β-$Ga_2O_3/(Al_xGa_{1-x})_2O_3$ quantum well structures without considering the impurity effect. It was found that the electronic transition wavelength can be tuned from shortwave infrared (1–3 μm) to far infrared (>30 μm) wavebands in this kind of low-dimensional structure. Their work gave us the impetus to further study the hydrogen-like shallow impurity states in β-$Ga_2O_3/(Al_xGa_{1-x})_2O_3$ core/shell nanostructures, where the electrons are confined in both the axial and radial directions. The binding energy as functions of structural dimension and aluminum composition when the impurity is located at different positions will be numerically computed using our previously developed algorithm that combines finite-difference approximation with a variational approach [11,12]. For comparison, both the hollow nanotube and solid nanorod core–shell structures, where the quantum confinement of electronic movement is differentiated in the core or shell layer, will be discussed in detail.

## 2. Theoretical Model

We studied a cylindrical core–shell nanostructure consisting of a $Ga_2O_3$ core layer and an $(Al_xGa_{1-x})_2O_3$ shell layer. The schematic of the core/shell nanostructure is given in Figure 1. Two types of this three-dimensional confined nanostructure, named nanorod and nanotube, were considered in our calculation. The radii of the core and shell layer of the nanorod are defined as $d_2$ and $d_3$, respectively. The nanostructure comprised a hollow structure to form a double-walled nanotube, and we defined the radius of the hollow region as $d_1$. The length of the nanorod or nanotube is defined as $L$. The z-axis is assumed to be along the nanostructure. Within the framework of effective mass approximation, the Hamiltonian of a conduction electron bound to a hydrogenic donor impurity shown in Figure 1b,c, bearing a charge $e$ located at $(\rho_0, \theta_0, z_0)$ in the $Ga_2O_3/(Al_xGa_{1-x})_2O_3$ core/shell nanostructure, is written in a cylindrical coordinate system as follows:

$$
\begin{aligned}
H = &-\frac{\hbar^2}{2m^*}\left(\frac{\partial^2}{\partial \rho^2} + \frac{1}{\rho}\frac{\partial}{\partial \rho} + \frac{1}{\rho^2}\frac{\partial^2}{\partial \theta^2}\right) - \frac{\hbar^2}{2m^*}\frac{\partial^2}{\partial z^2} + V_1(\rho) + V_2(z) \\
&- \frac{e^2}{\varepsilon_0 \varepsilon_r \sqrt{\rho^2 + \rho_0^2 - 2\rho\rho_0 \cos(\theta - \theta_0) + (z - z_0)^2}}
\end{aligned}
\tag{1}
$$

where $m^*$ is the electronic effective mass and $\varepsilon_r$ is the static dielectric constant.

For a nanorod, $V_1(\rho)$ is the radial confinement potential given by:

$$
V_1(\rho) = \begin{cases} 0, & 0 < \rho < d_2 \\ V_0, & d_2 \leq \rho \leq d_3, \\ \infty, & \rho > d_3 \end{cases}
\tag{2}
$$

where $V_0$ is the conduction band offset $\Delta E_c$ between $Ga_2O_3$ and $(Al_xGa_{1-x})_2O_3$. For a nanotube, $V_1(\rho)$ is written as

$$V_1(\rho) = \begin{cases} \infty, & 0 < \rho < d_1 \\ 0, & d_1 \leq \rho \leq d_2 \\ V_0, & d_2 < \rho \leq d_3 \\ \infty, & \rho > d_3 \end{cases}. \tag{3}$$

$V_2(z)$ is the axial confinement potential given by:

$$V_2(z) = \begin{cases} 0, & 0 \leq z \leq L \\ \infty, & z > L \end{cases}. \tag{4}$$

In order to obtain the ground state energy of the electron-impurity bound system, the variational approach is used to solve the coupled Schrödinger equation:

$$H\psi(\rho, z, \theta) = E\psi(\rho, z, \theta). \tag{5}$$

The two-parameter variational wave function is chosen as:

$$\psi(\rho, z, \theta) = Ce^{im\theta}\phi_l(\rho)\varphi_n(z)e^{-\alpha\sqrt{\rho^2+\rho_0^2-2\rho\rho_0\cos(\theta-\theta_0)}}e^{-\beta(z-z_0)}, \tag{6}$$

in which $C$ is the normalization constant of the wavefunction, and $\alpha$ and $\beta$ are the variational parameters that account for both in-plane and $z$-axial correlations between the electron and the impurity. $l$ and $n$ are the quantum numbers related to the radial and $z$-axial relative motion of an electron, respectively. The angular moment quantum number $m$ is taken as zero. The unbound electron states in the absence of impurities can be calculated using the method of separation of variables in the adiabatic approximation if the size difference of the core–shell nanostructure between the radial and axial directions is large. The boundary conditions at the interface and surface are determined by the Dirichlet boundary condition and continuity requirement. The radial wavefunction $\phi_l(\rho)$ and the $z$-axial wavefunction $\varphi_n(z)$ have the exact forms based on Bessel and trigonometric functions (e.g., Refs. [5–7]). In this work, we utilized our previously developed algorithm based on the finite difference method [14,15] to deal with both the radial and $z$-axial Schrödinger equations to obtain the wavefunctions and energy levels. Computation time can be saved without solving the transcendental equations. However, the numerical error mainly caused by boundary truncation of wavefunctions and differential segmentation is somewhat larger than that obtained by the algorithm using the exact solutions.

The radial Schrödinger equation is given as follows:

$$\left[-\frac{\hbar^2}{2m^*}\left(\frac{\partial^2}{\partial\rho^2} + \frac{1}{\rho}\frac{\partial}{\partial\rho}\right) + V_1(\rho)\right]\phi_l(\rho) = E_l\phi_l(\rho). \tag{7}$$

The radial Schrödinger equation in Equation (7) can be numerically solved using a finite-difference algorithm. First, the interval of $\rho$ in the radial direction is divided into $j$ parts, and thus the algebraic equations on the $j + 1$ nodes are solved simultaneously. The first-order central difference and second-order central difference formulas of the $k$-th node are given as:

$$\frac{d\phi_{l,k}(\rho)}{d\rho} = \frac{1}{2h}(\phi_{l,k+1} - \phi_{l,k-1}), \tag{8}$$

and

$$\frac{d^2\phi_{l,k}(\rho)}{d\rho^2} = \frac{1}{h^2}(\phi_{l,k+1} - 2\phi_{l,k} + \phi_{l,k-1}). \tag{9}$$

In Equations (8) and (9), $h$ is the step size. The position of the $k$-th node can be written as $\rho = hk$ and $V_1(\rho)$ can be expressed as $V_{1,k}$.

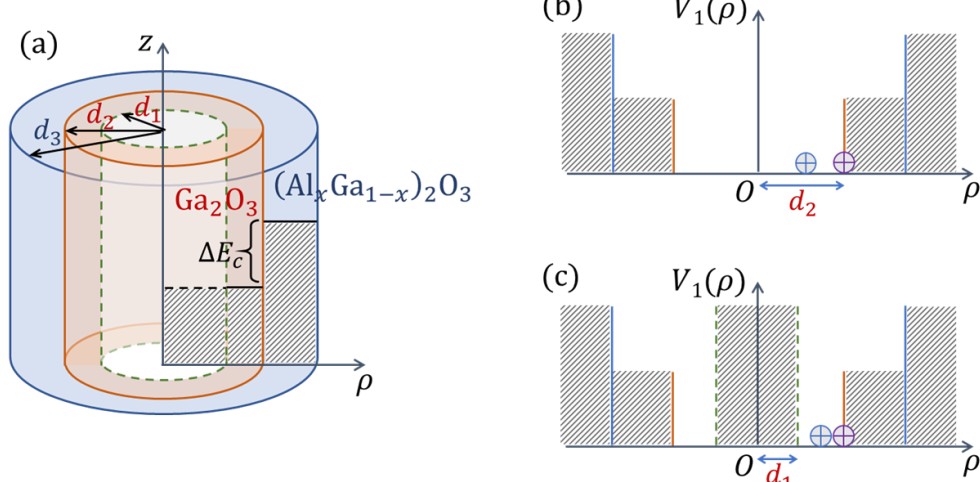

**Figure 1.** (**a**) Schematic representation of a β-Ga$_2$O$_3$/(Al$_x$Ga$_{1-x}$)$_2$O$_3$ core/shell nanostructure. (**b**) Schematic of the radial confinement potential of a core/shell nanorod. (**c**) Schematic of the radial confinement potential $V_1(\rho)$ of a core/shell nanotube. $d_1$, $d_2$, and $d_3$ denote the hollow tube radius, the core radius, and the shell radius. $\Delta E_c$ denotes the conduction band offset between Ga$_2$O$_3$ and (Al$_x$Ga$_{1-x}$)$_2$O$_3$. $O$ is the coordinate origin and '⊕' denotes the impurity located at different positions.

Therefore, the radial Schrödinger equation can be rewritten by substituting Equations (8) and (9) into Equation (7) in the finite-difference form, given as:

$$-\frac{\hbar^2}{2m^*}\left[\left(\frac{1}{h^2}-\frac{1}{2kh^2}\right)\phi_{l,k-1}-\frac{2}{h^2}\phi_{l,k}+\left(\frac{1}{h^2}+\frac{1}{2kh^2}\right)\phi_{l,k+1}\right]+V_{1,k}\phi_{l,k}=E_l\phi_{l,k}. \quad (10)$$

The algebraic equations from the first to the *j*-th node can be written as:

$$-\frac{\hbar^2}{2m^*}\left[\frac{1}{2h^2}\phi_{l,0}-\frac{2}{h^2}\phi_{l,1}+\frac{3}{2h^2}\phi_{l,2}\right]+V_{1,1}\phi_{l,1}=E_l\phi_{l,1}$$
$$-\frac{\hbar^2}{2m^*}\left[\frac{3}{4h^2}\phi_{l,1}-\frac{2}{h^2}\phi_{l,2}+\frac{5}{4h^2}\phi_{l,3}\right]+V_{1,2}\phi_{l,2}=E_l\phi_{l,2}$$

$$\cdots \cdots \cdots \cdots$$

$$-\frac{\hbar^2}{2m^*}\left[\frac{2j-3}{(2j-2)h^2}\phi_{l,j-1}-\frac{2}{h^2}\phi_{l,j}+\frac{2j-1}{(2j-2)h^2}\phi_{l,j+1}\right]+V_{1,j}\phi_{l,j}=E_l\phi_{l,j}. \quad (11)$$

The derivative of the wavefunction at $j+1$ equals 0, and the wavefunction reaches a maximum at the center of the core–shell nanostructure. So, the central boundary condition can be treated using the Newton interpolation method. It is obtained as follows:

$$-\frac{3}{2h}\phi_{l,0}+\frac{2}{h}\phi_{l,1}-\frac{1}{2h}\phi_{l,2}=0. \quad (12)$$

Equation (11) can then be written in matrix form as:

$$\frac{\hbar^2}{2m^*}\begin{pmatrix}\frac{4}{h^2} & -\frac{4}{h^2} & & & & \\ -\frac{1}{2h^2} & \frac{2}{h^2} & -\frac{3}{2h^2} & & & \\ & -\frac{3}{4h^2} & \frac{2}{h^2} & -\frac{5}{4h^2} & & \\ & \ddots & \ddots & \ddots & & \\ & & & -\frac{2j-3}{(2j-2)h^2} & \frac{2}{h^2} & -\frac{2j-1}{(2j-2)h^2} \\ & & & & -\frac{2j-1}{(2j-2)h^2} & \frac{2}{h^2}\end{pmatrix}\begin{pmatrix}\phi_{l,1} \\ \phi_{l,2} \\ \phi_{l,3} \\ \vdots \\ \phi_{l,j-1} \\ \phi_{l,j}\end{pmatrix}$$

$$+\begin{pmatrix} V_{1,1} & & & & & \\ & V_{1,2} & & & & \\ & & V_{1,3} & & & \\ & & & \ddots & & \\ & & & & V_{1,j-1} & \\ & & & & & V_{1,j} \end{pmatrix}\begin{pmatrix} \phi_{l,1} \\ \phi_{l,2} \\ \phi_{l,3} \\ \vdots \\ \phi_{l,j-1} \\ \phi_{l,j} \end{pmatrix} = E_l \begin{pmatrix} \phi_{l,1} \\ \phi_{l,2} \\ \phi_{l,3} \\ \vdots \\ \phi_{l,j-1} \\ \phi_{l,j} \end{pmatrix}. \tag{13}$$

By using the matrix transformation $D^{-1}CD = T$, in which:

$$D^{-1} = \begin{pmatrix} 1 & & & & & \\ & 2\sqrt{2} & & & & \\ & & 4 & & & \\ & & & 2\sqrt{6} & & \\ & & & & \ddots & \\ & & & & & 2\sqrt{2}\sqrt{j-1} \end{pmatrix}, \tag{14}$$

and

$$D = \begin{pmatrix} 1 & & & & & \\ & 1/(2\sqrt{2}) & & & & \\ & & 1/4 & & & \\ & & & 1/2\sqrt{6} & & \\ & & & & \ddots & \\ & & & & & 1/(2\sqrt{2}\sqrt{j-1}) \end{pmatrix}. \tag{15}$$

Equation (13) can be transformed as a $j \times j$ symmetric tridiagonal matrix

$$\frac{\hbar^2}{2m^*}\begin{pmatrix} \frac{4}{h^2} & -\frac{\sqrt{2}}{h^2} & & & & \\ -\frac{\sqrt{2}}{h^2} & \frac{2}{h^2} & -\frac{3\sqrt{2}}{4h^2} & & & \\ & -\frac{3\sqrt{2}}{4h^2} & \frac{2}{h^2} & -\frac{5\sqrt{3}}{6h^2} & & \\ & \ddots & \ddots & \ddots & & \\ & & & -\frac{2j-3}{\sqrt{2(j-3)2(j-1)}h^2} & \frac{2}{h^2} & -\frac{2j-1}{\sqrt{2(j-1)2j}h^2} \\ & & & & -\frac{2j-1}{\sqrt{2(j-1)2j}h^2} & \frac{2}{h^2} \end{pmatrix}\begin{pmatrix} \phi'_{l,1} \\ \phi'_{l,2} \\ \phi'_{l,3} \\ \vdots \\ \phi'_{l,j-1} \\ \phi'_{l,j} \end{pmatrix}$$

$$+\begin{pmatrix} V_{1,1} & & & & & \\ & V_{1,2} & & & & \\ & & V_{1,3} & & & \\ & & & \ddots & & \\ & & & & V_{1,j-1} & \\ & & & & & V_{1,j} \end{pmatrix}\begin{pmatrix} \phi'_{l,1} \\ \phi'_{l,2} \\ \phi'_{l,3} \\ \vdots \\ \phi'_{l,j-1} \\ \phi'_{l,j} \end{pmatrix} = E_l \begin{pmatrix} \phi'_{l,1} \\ \phi'_{l,2} \\ \phi'_{l,3} \\ \vdots \\ \phi'_{l,j-1} \\ \phi'_{l,j} \end{pmatrix}. \tag{16}$$

Finally, the energy level $E_l$ can be obtained by solving the minimum eigenvalues and eigenvectors of the $j \times j$ symmetric tridiagonal matrix, and the radial wavefuntion $\phi_l(\rho)$ can be obtained by another matrix transformation:

$$\begin{pmatrix} \phi_{l,1} \\ \phi_{l,2} \\ \phi_{l,3} \\ \vdots \\ \phi_{l,j-1} \\ \phi_{l,j} \end{pmatrix} = \begin{pmatrix} 1 & & & & & \\ & 1/(2\sqrt{2}) & & & & \\ & & 1/4 & & & \\ & & & 1/(2\sqrt{6}) & & \\ & & & & \ddots & \\ & & & & & 1/(2\sqrt{2}\sqrt{j-1}) \end{pmatrix}\begin{pmatrix} \phi'_{l,1} \\ \phi'_{l,2} \\ \phi'_{l,3} \\ \vdots \\ \phi'_{l,j-1} \\ \phi'_{l,j} \end{pmatrix}. \tag{17}$$

The *z*-axial Schrödinger equation is written as follows:

$$\left[-\frac{\hbar^2}{2m^*}\frac{\partial^2}{\partial z^2} + V_2(z)\right]\varphi_n(z) = E_n\varphi_n(z). \tag{18}$$

It can be solved using another finite-difference algorithm, which is somewhat different from that used to solve the radial Schrödinger equation. The interval of *z* in the axial direction is divided into *i* parts, and thus the algebraic equations on the *i* + 1 nodes are solved simultaneously. The first-order central difference and second-order central difference formulas of the *k*-th node are given as:

$$\frac{d\varphi_{n,k}(z)}{dz} = \frac{1}{2h}(\varphi_{n,k+1} - \varphi_{n,k-1}), \tag{19}$$

$$\frac{d^2\varphi_{n,k}(z)}{dz^2} = \frac{1}{h^2}(\varphi_{n,k+1} - 2\varphi_{n,k} + \varphi_{n,k-1}). \tag{20}$$

Therefore, the axial Schrödinger equation can be rewritten by substituting Equations (19) and (20) into Equation (18) in the finite-difference form, given as:

$$-\frac{\hbar^2}{2m^*}\left[\frac{1}{h^2}\varphi_{n,k-1} - \frac{2}{h^2}\varphi_{n,k} + \frac{1}{h^2}\varphi_{n,k+1}\right] + V_{2,k}\varphi_{n,k} = E_n\varphi_{n,k}, \tag{21}$$

in which $z = hk$ and $V_2(z)$ can be expressed as $V_{2,k}$. The boundary condition is very different from the radial Schrödinger equation. Since the outmost region of a core–shell nanostructure is assumed to be a vacuum, the *z*-axial wavefunction reaches 0 at the boundary, that is, $\varphi_{n,1} = 0$ and $\varphi_{n,i+1} = 0$.

Thus, the algebraic equations from the second to the *i*-th node can be written as:

$$-\frac{\hbar^2}{2m^*}\left[-\frac{2}{h^2}\varphi_{n,2} + \frac{1}{h^2}\varphi_{n,3}\right] + V_{2,2}\varphi_{n,2} = E_n\varphi_{n,2}$$
$$-\frac{\hbar^2}{2m^*}\left[\frac{1}{h^2}\varphi_{n,2} - \frac{2}{h^2}\varphi_{n,3} + \frac{1}{h^2}\varphi_{n,4}\right] + V_{2,3}\varphi_{n,3} = E_n\varphi_{n,3}$$
$$-\frac{\hbar^2}{2m^*}\left[\frac{1}{h^2}\varphi_{n,3} - \frac{2}{h^2}\varphi_{n,4} + \frac{1}{h^2}\varphi_{n,5}\right] + V_{2,4}\varphi_{n,4} = E_n\varphi_{n,4}$$
$$\cdots \cdots \cdots \cdots$$
$$-\frac{\hbar^2}{2m^*}\left[\frac{1}{h^2}\varphi_{n,i-1} - \frac{2}{h^2}\varphi_{n,i}\right] + V_{2,i}\varphi_{n,i} = E_n\varphi_{n,i}. \tag{22}$$

Equation (22) can then be written in matrix form as:

$$\frac{\hbar^2}{2m^*}\begin{pmatrix} \frac{2}{h^2} & -\frac{1}{h^2} & & & & \\ -\frac{1}{h^2} & \frac{2}{h^2} & -\frac{1}{h^2} & & & \\ & -\frac{1}{h^2} & \frac{2}{h^2} & -\frac{1}{h^2} & & \\ & \ddots & \ddots & \ddots & & \\ & & & -\frac{1}{h^2} & \frac{2}{h^2} & -\frac{1}{h^2} \\ & & & & -\frac{1}{h^2} & \frac{2}{h^2} \end{pmatrix}\begin{pmatrix} \varphi_{n,2} \\ \varphi_{n,3} \\ \varphi_{n,4} \\ \vdots \\ \varphi_{n,i-1} \\ \varphi_{n,i} \end{pmatrix}$$

$$+\begin{pmatrix} V_{2,2} & & & & & \\ & V_{2,3} & & & & \\ & & V_{2,4} & & & \\ & & & \ddots & & \\ & & & & V_{2,i-1} & \\ & & & & & V_{2,i} \end{pmatrix}\begin{pmatrix} \varphi_{n,2} \\ \varphi_{n,3} \\ \varphi_{n,4} \\ \vdots \\ \varphi_{n,i-1} \\ \varphi_{n,i} \end{pmatrix} = E_n\begin{pmatrix} \varphi_{n,2} \\ \varphi_{n,3} \\ \varphi_{n,4} \\ \vdots \\ \varphi_{n,i-1} \\ \varphi_{n,i} \end{pmatrix}. \tag{23}$$

Because the matrix in the above equation itself is an $(i - 1) \times (i - 1)$ symmetric tridiagonal matrix, the energy level $E_n$ and *z*-axial wavefuntion $\varphi_n(z)$ can be directly obtained by solving the minimum eigenvalues and eigenvectors of the $(i - 1) \times (i - 1)$ symmetric tridiagonal matrix without a matrix transformation.

Finally, the binding energy of an impurity can be solved by:

$$E_{\mathrm{b}} = E_l + E_n - E_{\mathrm{D}}, \tag{24}$$

where the expectation energy $E_{\mathrm{D}}$ for a certain impurity state $(l, n, m)$ can be derived by the energy minimization:

$$E_{\mathrm{D}} = \min_{\alpha,\beta}\langle\psi(\rho,z,\theta)|\widehat{H}|\psi(\rho,z,\theta)\rangle. \tag{25}$$

In the literature, most authors only calculated the binding energy of the impurity ground state with $l = 0$, $n = 0$. However, there are many states related to the quantum numbers $l$ and $n$; we mainly take the lowest two states, i.e., $l = 0$ or 1 and $n = 0$ or 1, into account for simplicity. In other words, if we assume $m = 0$, four impurity states denoted as $\Psi_{00}$, $\Psi_{01}$, $\Psi_{10}$, and $\Psi_{11}$ are considered in our computation.

## 3. Numerical Results and Discussion

The related parameters of $\beta$-$Ga_2O_3$ and $\beta$-$(Al_xGa_{1-x})_2O_3$ used in the computation are listed in Table 1. According to the references, the calculation was performed by only considering the difference in the effective mass between the inner layer $Ga_2O_3$ and the barrier material $(Al_xGa_{1-x})_2O_3$. The conduction band offset between $Ga_2O_3$ and $(Al_xGa_{1-x})_2O_3$ were taken as $2.15x + 0.94x^2$ eV for $x < 0.5$ and $0.96–0.24x + 1.95x^2$ eV for $x > 0.5$. The dielectric constants of $Ga_2O_3$ and $(Al_xGa_{1-x})_2O_3$ were assumed to be 10. Without loss of generality, the length $L$ of both the nanorod and nanotube was fixed at $L = 30$ nm and the radius of the outer shell layer was $d_3 = 20$ nm. To obtain the binding energy of the impurity states, we first calculated the wavefunctions of the impurity-bound electron states.

**Table 1.** Material parameters used in the computation [31]. Copyright 2020, AIP Publishing.

| Material parameters | $\beta$-$Ga_2O_3$ | $\beta$-$(Al_xGa_{1-x})_2O_3$ |
|---|---|---|
| Effective mass $m^*$ ($m_0$) | 0.28 | $0.28 + 0.11x$ |
| Band gap $E_g$ (eV) | 4.69 | $4.69 + 1.34x + x^2$ |
| Dielectric constant $\varepsilon_r$ ($\varepsilon_0$) | 10 | 10 |

Figure 2 illustrates the probability density in the $\rho$-$z$ plane of the impurity states corresponding to $\Psi_{00}$, $\Psi_{01}$, $\Psi_{10}$, and $\Psi_{11}$ in a $\beta$-$Ga_2O_3/(Al_xGa_{1-x})_2O_3$ core/shell nanorod and in a $\beta$-$Ga_2O_3/(Al_xGa_{1-x})_2O_3$ core/shell nanotube. The impurity was assumed to be located at the position $(d_2/2, L/2)$ in the $\rho$-$z$ plane in the nanorod and at the position $(d_2/2 + d_1/2, L/2)$ in the nanotube, respectively. It is obvious that the distribution of the probability density features symmetry regarding $z = L/2$ (15 nm) for all of the impurity states in the two nanostructures. It also implies that the probability density has a circular symmetry for the four impurity states in the two kinds of nanostructures due to the structural geometry. As for the nanorod structure, the probability density related to $\Psi_{00}$ is mainly distributed around the impurity and spreads over the core $Ga_2O_3$ layer. There are two or more distribution regions of the probability density for the excited states. The two parts of the probability density related to $\Psi_{01}$ are distributed close to the core center, while the main probability density related to $\Psi_{10}$ goes toward the interface, but this impurity state has a small part distributed near the core center. Most of the probability density of $\Psi_{11}$ is distributed in the core region, but some is near the interface. As for the nanotube structure, the situation is quite different. The probability density concerning $\Psi_{00}$ is mainly distributed around the impurity and closer to the interface between $Ga_2O_3$ and $(Al_xGa_{1-x})_2O_3$. The probability density related to the three excited impurity states moves much closer to the interface, except $\Psi_{01}$. The probability density related to $\Psi_{01}$ is located more uniformly in the inner $Ga_2O_3$ layer. It is concluded that if the quantum number $l > n$, the probability densities of the impurity states change more prominently since the finite confinement potential at the $\rho$-axis is weaker than the confinement potential at the $z$-axis, which can be found in Equations (3) and (4). We also mention that tunneling behavior hardly occurs in

the two nanostructures due to the large width of the confined core region as well as the impurity Coulombic interaction.

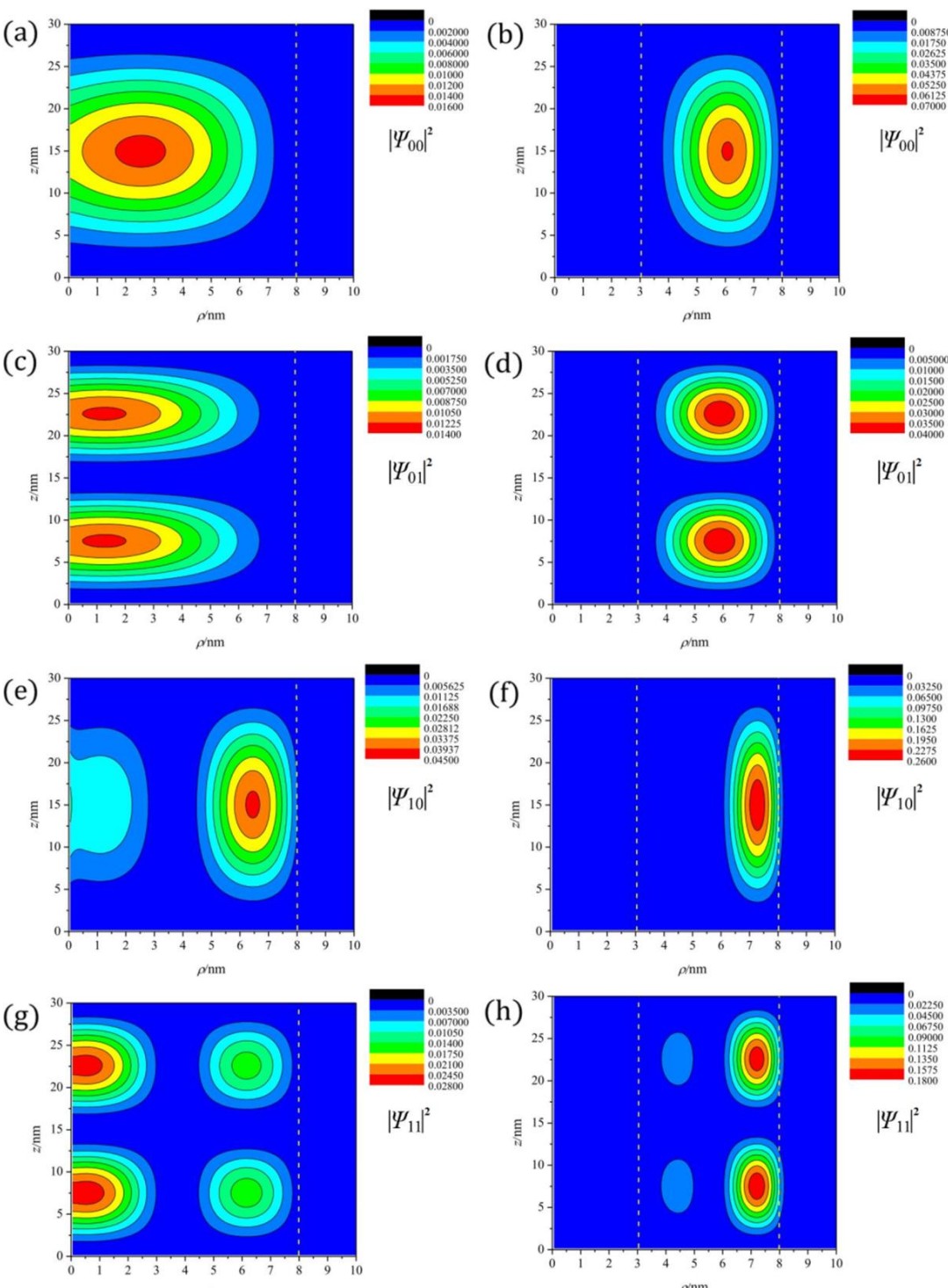

**Figure 2.** Probability densities in the $\rho$-$z$ plane of the impurity states (**a**) $\Psi_{00}$, (**c**) $\Psi_{01}$, (**e**) $\Psi_{10}$, and (**g**) $\Psi_{11}$ in a $\beta$-$Ga_2O_3/(Al_{0.3}Ga_{0.7})_2O_3$ core/shell nanorod with $d_2 = 8$ nm where the impurity is located at the position $(d_2/2, L/2)$, and the impurity states (**b**) $\Psi_{00}$, (**d**) $\Psi_{01}$, (**f**) $\Psi_{10}$, and (**h**) $\Psi_{11}$ in a $\beta$-$Ga_2O_3/(Al_{0.3}Ga_{0.7})_2O_3$ core/shell nanotube with $d_1 = 3$ nm and $d_2 = 8$ nm where the impurity is located at the position $(d_2/2 + d_1/2, L/2)$. Note that only a half of the cross-section is plotted because of the symmetry.

Figure 3 shows the change in impurity binding energy $E_b$ with increasing core radius $d_2$ for a core/shell nanorod and increasing hollow tube radius $d_1$ for a core/shell nanotube. The order in terms of the magnitude of $E_b$ for different impurity states is $\Psi_{00} < \Psi_{01} < \Psi_{10} < \Psi_{11}$ in both nanostructures. It can be seen from Figure 3a that $E_b$ decreases as the core radius $d_2$ increases from 2 to 7 nm, which can be attributed to the weaker quantum confinement. This variation behavior is very similar to that in Ref. [12]. The influence of the impurity position (at the interface or in the core region) on $E_b$ does not seem to be very prominent. It has a stronger impact on the binding energy corresponding to the ground state, $\Psi_{00}$. The binding energy when the impurity is at the interface is lower than that when the impurity appears in the core region. If the impurity is not at the center $\rho = 0$, the trend in the binding energy of the ground state goes toward zero. If the impurity is at the center, the binding energy will reach the bulk value. In the contrast, one can see from Figure 3b that $E_b$ increases as the hollow tube radius $d_1$ of the core/shell nanotube increases. In other words, the inner layer of the double walls of the nanotube become narrower, thus enhancing the quantum confinement of the impurity states. The impurity position has a more obvious influence on the binding energy, especially related to the excited impurity states, in a nanotube than in a nanorod. The difference in $E_b$ between two impurity positions is enhanced when $d_1$ is larger. In the extreme case, the ground state binding energy will reach zero if $d_1$ goes to zero. It needs to be pointed out that the binding energy corresponding to the two excited states, $\Psi_{10}$ and $\Psi_{11}$, decreases when the impurity moves from the inner layer to the interface between the two walls of the nanotube.

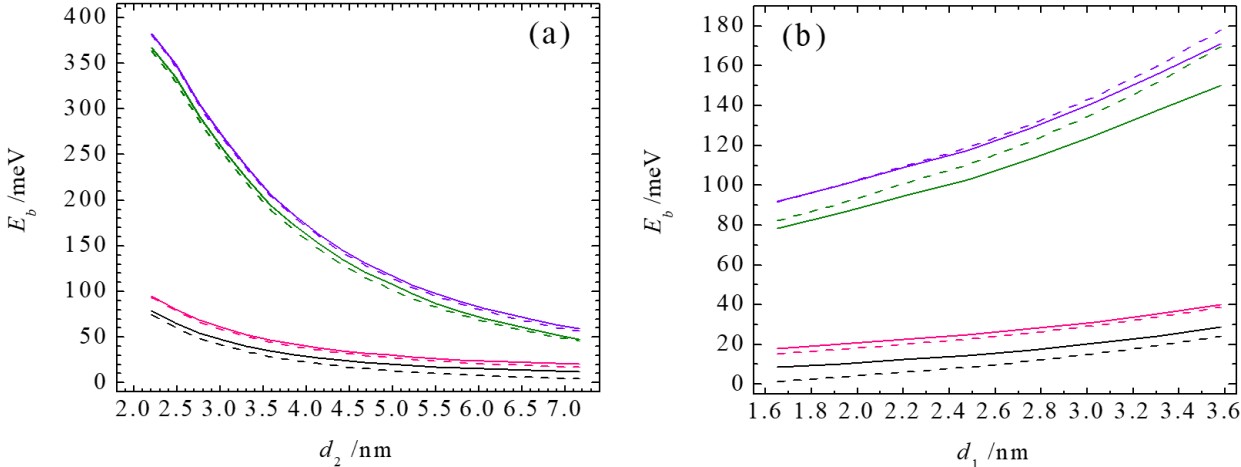

**Figure 3.** Impurity binding energy $E_b$ as a function of (**a**) core radius $d_2$ for a β-Ga$_2$O$_3$/(Al$_{0.3}$Ga$_{0.7}$)$_2$O$_3$ core/shell nanorod and (**b**) hollow tube radius $d_1$ for a β-Ga$_2$O$_3$/(Al$_{0.3}$Ga$_{0.7}$)$_2$O$_3$ core/shell nanotube. The black, pink, green, and purple lines denote the contributions of the impurity states $\Psi_{00}$, $\Psi_{01}$, $\Psi_{10}$, and $\Psi_{11}$, respectively. The solid and dashed lines denote the impurity located at the position $(d_2/2, L/2)$ and $(d_2, L/2)$ for a nanorod and at the position $(d_2/2 + d_1/2, L/2)$ and $(d_2, L/2)$ for a nanotube, respectively.

Next, we turn to discuss the influence of Al composition on the binding energy of the impurity located at different positions in the two kinds of nanostructures. It can be observed from Figure 4 that an inflection appears especially for the excited impurity states, which can be attributed to the conduction band offset between Ga$_2$O$_3$ and (Al$_x$Ga$_{1-x}$)$_2$O$_3$ at $x = 0.5$ [22]. As the Al composition increases, the binding energy also increases because the probability density becomes more located in the inner layer, which is confined by a higher potential barrier. This increment is less significant for the $\Psi_{00}$ and $\Psi_{01}$ states. Moreover, the binding energy decreases if the impurity moves towards the interface. The binding energy for the $\Psi_{10}$ and $\Psi_{11}$ states in a nanotube structure is somewhat different, which can be also seen in Figure 3b, possibly because of the quantum tunneling towards the (Al$_x$Ga$_{1-x}$)$_2$O$_3$ barrier layer. If the radial dimension is smaller than the electronic mean free path, the

quantum tunneling at the tubular interface will make a greater contribution to the binding energy of the off-center hydrogen-like impurity in a core/shell nanostructure. However, the phase transition from β to α phase and the strain effect, which may both alter energy band diagrams at high compositions approaching $x = 0.7$, were not taken into account in this paper. These will be carefully considered in future work. We point out that degenerate states with different radial and axial quantum numbers, which were always neglected in most of the literature, should be considered in some quantum confined structures like rods, tubes, rings, ribbons, disks, and so forth.

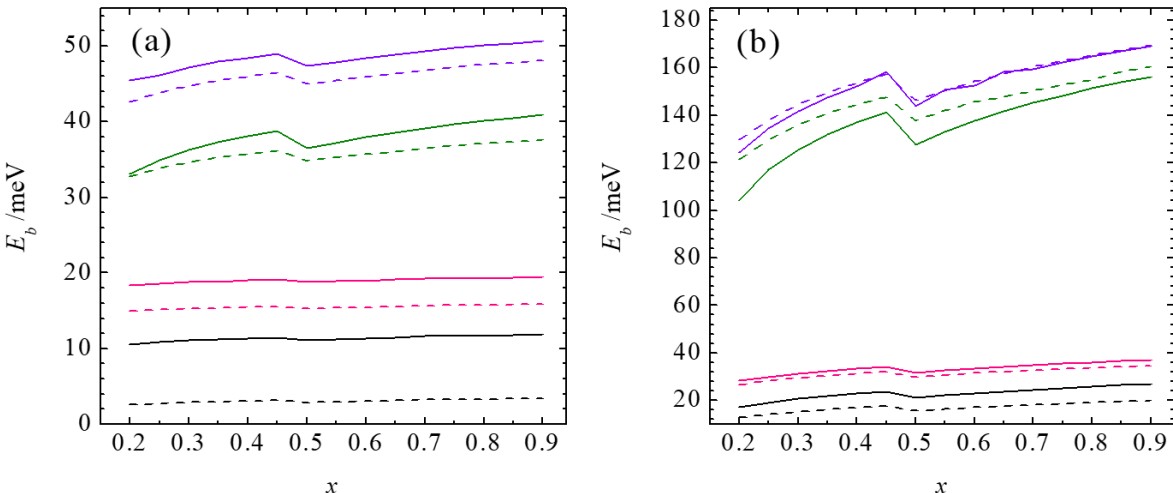

**Figure 4.** Impurity binding energy $E_b$ as a function of the aluminum composition $x$ for a β-$Ga_2O_3$/$(Al_xGa_{1-x})_2O_3$ core/shell (**a**) nanorod with $d_2$ = 8 nm and (**b**) nanotube with $d_1$ = 3 nm and $d_2$ = 8 nm. The black, pink, green, and purple lines denote the contributions of the impurity states $\Psi_{00}$, $\Psi_{01}$, $\Psi_{10}$, and $\Psi_{11}$, respectively. The solid and dashed lines denote the impurity located at the position $(d_2/2, L/2)$ and $(d_2, L/2)$ for a nanorod and at the position $(d_2/2 + d_1/2, L/2)$ and $(d_2, L/2)$ for a nanotube, respectively.

## 4. Conclusions

In conclusion, we have made a comparison of the impurity states in a coaxial core/shell nanorod and coaxial core/shell nanotube composed of the β-$Ga_2O_3$/$(Al_xGa_{1-x})_2O_3$ material system. The probability density and binding energy of an off-center hydrogenic donor impurity with regards to four impurity states, namely $\Psi_{00}$, $\Psi_{01}$, $\Psi_{10}$, and $\Psi_{11}$, were numerically calculated using the finite-difference method combined with a variational approach in the framework of effective mass approximation and single electron approximation. Due to the synergic mechanism of nanostructured quantum confinement and Coulombic interaction from the impurity, the quantum localization maps of the four impurity-bound electron states were quite different for core/shell nanorods and nanotubes. As a consequence, varying the Al composition and geometrical parameters makes it possible to modify the density of the electron states and the binding energy of impurities in these kinds of core/shell nanostructures. Comparatively, the impurity binding energies corresponding to the excited states were larger than the ground state binding energy and more sensitive to the structural dimension and Al composition. If the radial dimension is very small (smaller than the electronic mean free path), the interface impurity plays a more important role in determining the electronic properties due to the quantum tunneling at the tubular interface, especially in a nanotube structure. It is believed that our results will help to measure the electrical transport and electroluminescence properties of rapidly developing $Ga_2O_3$/$(Al_xGa_{1-x})_2O_3$ low-dimensional nanostructures for device applications.

**Author Contributions:** S.H.: Methodology, Software, Data curation, Visualization, Validation, Writing—review & editing. J.Z.: Conceptualization, Writing-original draft, Supervision, Investigation, Writing—review & editing. All authors have read and agreed to the published version of the manuscript.

**Funding:** This research was funded by the National Natural Science Foundation of China (Grant No. 12164031) and the Science Foundation of Inner Mongolia Autonomous Region (Grant No. 2020MS06007 and 2023LHMS01004). The APC was funded by the National Natural Science Foundation of China (Grant No. 12164031).

**Data Availability Statement:** The data that support the findings of this study are available within the article.

**Conflicts of Interest:** The authors declare no conflict of interest.

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
