# Peer review of "Hydrogen-like Impurity States in β-Ga2O3/(AlxGa1−x)2O3 Core/Shell Nanostructures: Comparison between Nanorods and Nanotubes"

_crystals, doi:10.3390/cryst13081227_

Round 1

Reviewer 1 Report

Hydrogen-like impurity states in β-Ga2O3/(AlxGa1-x)2O3 2 core/shell nanostructures: Comparison between nanorods and nanotubes is very interesting paper. Some remarks are required!

Line 2: Hydrogen-like impurity (which type of impurities are expected)

Line 10: β-Ga2O3/(AlxGa1-x)2O3 core/shell nanostructure (prepared by method)

Line 128: but this impurity states (which type of impurities)

Line 156: Fig. 3 shows the change of impurity binding energy Eb with increasing the core radius (what compound is core) for a core/shell (which compound is shell) nanorod

Line 160: the core radius d2 increases (in which range)

Line 171: excited impurity states (which type of impurities are present? How are these impurities reached core (shell structure?)

Line 171: What are dimensions of in a nanotube than in a nanorod? Did you use same dimension for comparison?

line 211 hydrogenic donor  impurity (please to write one type of hydrogenic donor)

Line 218: the geometrical parameters (what are geometrical parameters)

Line 222: If the radial dimension is very small (in which range?)

Author Response

Line 2: Hydrogen-like impurity (which type of impurities are expected)

In the single-band effective mass model, the impurity is generally simplified as a positive charge center. As far as we know, the shallow impurities used for effective n-type doping in Ga2O3 is H, Si, Ge, Sn and so on. The impurities can also be unintentional dopants during the growth process and the point defects.

Line 10: β-Ga2O3/(AlxGa1-x)2O3 core/shell nanostructure (prepared by method)

There is very little works related β-Ga2O3/(AlxGa1-x)2O3 low-dimensional structures especially core/shell nanostructures (All the corresponding references which we can retrieve are listed in our revised manuscript). According to the ref. 30 in the revised manuscript, the Ga2O3@Al2O3 core−shell nanowires were prepared by the plasma-enhanced atomic layer deposition.

Line 128: but this impurity states (which type of impurities)

We corrected the word “states” as “state” in the revised manuscript and this impurity state refers to Ψ10. We inserted Ψ00, Ψ01, Ψ10 and Ψ11 in figure 3 to make it clearer.

Line 156: Fig. 3 shows the change of impurity binding energy Eb with increasing the core radius (what compound is core) for a core/shell (which compound is shell) nanorod

We rewrote the figure captions to make it clear.

Line 160: the core radius d2 increases (in which range)

The core radius d2 increases from 2 nm to 7 nm which was noted in the revised manuscript.

Line 171: excited impurity states (which type of impurities are present? How are these impurities reached core (shell structure?)

The impurity position was assumed in the numerical simulation. The excited impurity states are the states with the quantum numbers exceeding 1.

Line 171: What are dimensions of in a nanotube than in a nanorod? Did you use same dimension for comparison?

The width of the shell layer (AlxGa1-x)2O3 of the nanorod and nanotube was taken as the same for comparison.

line 211 hydrogenic donor  impurity (please to write one type of hydrogenic donor)

As far as we know, the shallow impurities used for effective n-type doping in Ga2O3 is H, Si, Ge, Sn and so on. The impurities can also be unintentional dopants during the growth process and the point defects.

Line 218: the geometrical parameters (what are geometrical parameters)

The geometrical parameters are the thickness of the core and shell layers. Al composition mainly influence the potential barrier height of electron confinement.

Line 222: If the radial dimension is very small (in which range?)

The radial dimension is smaller than electronic mean free path ~2.5 nm (Thin Solid Films 2009, 517, 1928).

Reviewer 2 Report

The paper is devoted to the theoretical study of hydrogen impurity states in β-Ga2O3/(AlxGa1-x)2O3 nanostructures of core/shell type for the cases of nanorods and nanotubes. In this paper, a theoretical model for the cases of nanorods and nanotubes is presented. Numerical simulations of the two cases are carried out. For nanorods, the probability density is mainly distributed around the impurity and spreads over the Ga2O3 core layer. For the nanotube structure, the situation is quite different. The probability density is mainly distributed around the impurity and closer to the Ga2O3 interface. Also in this work, the influence of Al composition in (AlxGa1-x)2O3 on the binding energy of impurities located in different positions in the two types of nanostructures is investigated.

There are no significant remarks on the article. The only thing to note is that the case of β-Ga2O3/(AlxGa1-x)2O3 nanotubes is hypothetical and not realizable in the near future. Therefore, it is reasonable to simply dwell on nanorods with a shell, where in fact there are also problems with the formation of such structures.

In terms of content, the article does not quite correspond to the journal Crystals, which should present the results on materials preparation and investigation of their structural and physical properties. Therefore, it may be appropriate to redirect the article to another journal.

Remarks:

(1) In Figures 3-4, it is useful to insert Ψ00, Ψ01, Ψ10 and Ψ11 for dependencies in the appropriate color for accessibility of information. There is an explanation in the figure caption, but it is better to insert directly into the figures.

There are a number of remarks of a technical character:

(1) line 78: the beginning of the line without indentation

(2) a comma is needed after formula (2)

(3) line 80: beginning of line without indentation

(4) line 78: beginning of line without indentation

(5) formula (9) must be followed by a comma

(6) line 112: beginning of line without indentation

After the remarks are corrected, the article can be published.

English is correct.

Author Response

We inserted the Ψ00, Ψ01, Ψ10 and Ψ11 in Figure 3 and rewrote all the figure captions to make them clearer in the revised manuscript. The remarks of a technical character were corrected in the revised manuscript.